# The Association between Endometriosis and Obstructive Müllerian Anomalies

**DOI:** 10.3390/biomedicines12030651

**Published:** 2024-03-14

**Authors:** Nozomi Takahashi, Miyuki Harada, Mayuko Kanatani, Osamu Wada-Hiraike, Yasushi Hirota, Yutaka Osuga

**Affiliations:** Department of Obstetrics and Gynecology, Faculty of Medicine, The University of Tokyo, 7-3-1, Hongo, Bunkyo, Tokyo 113-8655, Japan

**Keywords:** cervical aplasia, dysmenorrhea, endometriosis, Herlyn–Werner–Wunderlich syndrome, Müllerian anomalies

## Abstract

It is unclear whether clinical background differs between endometriosis in adolescent patients with obstructive Müllerian anomalies and those without anomalies. The aim of the study is to identify the difference in clinical characteristics of endometriosis in patients with or without obstructive Müllerian anomalies. The study involved 12 patients aged under 24 years old who underwent primary surgery for obstructive Müllerian anomalies and 31 patients aged under 24 years old who underwent surgery for ovarian endometrioma. A total of 6 out of 12 cases with obstructive Müllerian anomalies developed endometriosis (4 Herlyn–Werner–Wunderlich syndrome, 2 non-communicating functional uterine horn, 2 cervical aplasia). The age at surgery was significantly younger in endometriosis with obstructive Müllerian anomalies, compared with those without obstructive Müllerian anomalies (17.8 ± 4.4 vs. 23.1 ± 1.3, *p* = 0.0007). The rate of endometrioma was 50.0% and the rate of hydrosalpinx was significantly higher (66.7% vs. 0%, *p* = 0.0002) in the group of obstructive Müllerian anomalies. The recurrence rate of endometriosis was 20.0% in the group of anomalies and 25.9% in the group of those without anomalies. Adolescent patients with obstructive Müllerian anomalies more easily developed endometriosis and co-occurred with higher rate of hematosalipinx.

## 1. Introduction

Congenital malformations of the female genital tract have a variety of morphologies that include vaginal closure, cervical closure, noncommunicating functional uterine horn, and Herlyn–Werner–Wunderlich (HWW) syndrome. HWW syndrome is described as a syndrome consisting of uterus didelphys, unilateral renal aplasia, and ipsilateral blind hemivagina [1], which is also known as obstructed hemivagina and ipsilateral renal agenesis (OHVIRA). Vaginal and cervical closure causes primary amenorrhea, but in the noncommunicating functional uterine horn and HWW syndrome, obstruction of the menstrual outflow typically causes a cyclic or recurrent pelvic pain.

Endometriosis is defined as the presence of endometrial tissue exterior to the uterus such as in the ovary and the peritoneal cavity. One risk factor for endometriosis is obstruction of menstrual outflow [2]; therefore, it is possible that obstructive malformations of the female genital tract increase the risk of development of endometriosis. Most patients with endometriosis diagnosed by surgery are aged 25 years old and over. The nationwide cohort study reported that the prevalence of surgically verified endometriosis among young patients (10–19 years old) was 1.4% in 26,301 women [3]. Systematic review showed that 648 out of 1011 (64%) symptomatic adolescents undergoing laparoscopy were found to have endometriosis [4]. However, the etiology and mechanisms of the early onset of endometriosis, especially in adolescence, is not well known.

It is reported that endometriosis in young girls tends to be co-occurring with uterine malformations. Matalliotakis et al. reported that the congenital structural abnormality of the reproductive organs was observed much more in the younger cases (16/55, 32%), compared with the older cases (14/485, 1.6%) [5]. Another study reported that endometriosis was diagnosed in 23/50 (46%) of patients with obstructive Müllerian anomalies [6]. On the other hand, patients without Müllerian anomalies can also develop endometriosis at young ages. Retrograde menstruation cannot explain all the mechanisms of the development of endometriosis, and the etiology of endometriosis in adolescents may be different from that of adult patients [7]. However, information about the difference between endometriosis in adolescent patients with obstructive Müllerian anomalies and those without anomalies is limited.

The aim of this study is to assess the development of endometriosis in adolescence with obstructive Müllerian anomalies and detect the difference in clinical characteristics between endometriosis patients with or without Müllerian anomalies.

## 2. Materials and Methods

### 2.1. Study Design

This study is a retrospective cohort study. The inclusion criteria of this study are as follows: (1) patients aged under 24 years old who underwent primary surgery for obstructive Müllerian anomalies, and (2) patients aged under 24 years old who underwent surgery for ovarian endometrioma between January 2008 and October 2017. All patients with obstructive Müllerian anomalies underwent surgery in our hospital. The indication of surgery for ovarian endometrioma is that the size is over 6 cm. Two patients with a previous operation history were excluded. Among 131 patients aged under 24 years old who underwent surgeries for benign gynecological diseases in this period, 43 patients in total were included in this study. The average age was 22.0 ± 2.7 (mean ± SD), and 7 patients were under 19 years old. In 12 cases, surgery was performed for obstructive Mullerian anomalies, and 31 patients underwent surgery for ovarian endometrioma. The surgery was performed in the Division of Obstetrics and Gynecology at the University of Tokyo Hospital. For the patients with obstructive genital malformation suffering from hematometra, hysteroscopic hemivaginal septal resection or laparoscopic hysterectomy was performed. For the endometrioma patients, laparoscopic cystectomy was performed. The patients with a previous operation history were excluded.

### 2.2. Data Collection

Patients’ medical records were retrospectively studied until February 2020. The following data were assessed: age at menarche, age at surgery, type of surgery, interval between first symptom or menarche to a surgery, revised American Society for Reproductive Medicine (rASRM) classification, the presence of endometrioma or hematosalpinx, postoperative hormonal treatment, follow-up period, and the recurrence of endometriosis. Preoperative evaluation of each patient involved clinical examination, pelvic and abdominal ultrasound (US), and pelvic magnetic resonance imaging (MRI). HWW syndrome was diagnosed with the presence of renal agenesis, ipsilatarel hemivagina, and didelphys uterus [1]. The presence of endometriosis was diagnosed based on rASRM classification [8], endometrioma was confirmed by pathology, and hematosalpinx was confirmed based on laparoscopic findings. Patients were postoperatively followed up every 3 months and underwent pelvic exam and US to examine the recurrence of endometrioma.

This retrospective study was approved by the institutional review board of The University of Tokyo. All patients were informed that the patients’ clinical data may be used for research and scientific publications, and we obtained signed informed consent prior to surgery in each patient.

### 2.3. Statistical Analysis

Statistical analysis was performed using JMP Pro 14 software (SAS Institute Inc., Cary, NC, USA) and GraphPad Prism 10 (GraphPad Software Inc., Boston, MA, USA). The Shapiro–Wilk test was used for a test of normal distribution. All continuous data was not normally distributed. Patients’ characteristics between patients with or without obstructive Müllerian anomaly were compared using Fisher’s exact test for proportions and the Mann–Whitney test for continuous variables. A *p*-value of less than 0.05 was considered statistically significant, and all reported *p*-values were one-sided.

## 3. Results

### 3.1. The Diagnosis and Surgical Management of Patients with Obstructive Müllerian Anomalies

Table 1 lists the diagnosis and the primary surgery for congenital structural abnormalities with obstructive Müllerian anomalies. Out of 12 cases, 8 patients were diagnosed with HWW syndrome, and 2 patients had cervical abnormality. Six patients with typical HWW syndrome without cervical abnormality had hematocolpos, and they underwent vaginal septal resection without laparoscopic inspection. Two of them underwent hysteroscopic examination. Therefore, it could not be known whether endometriosis was co-occurring exactly, but there were no cases in which endometrioma or hematosalpinx was observed on the MRI. Overall, five out of six (83.3%) patients had dysmenorrhea before surgery.

Out of 12 cases, 6 cases were co-occurring with endometriosis. Two cases of HWW syndrome with unilateral cervical hypoplasia underwent laparoscopic unilateral supra-cervical hysterectomy. Two cases of non-communicating functional uterine rudimentary horn had hematosalpinx on the abnormal uterine side, and we performed laparoscopic uterine horn removal and unilateral tubectomy. Two cases had cervical aplasia and upper vaginal defect. One of them underwent laparoscopic plastic surgery to vagina, and another case underwent diagnostic laparoscopy.

### 3.2. Characteristics of Endometriosis Patients with Obstructive Müllerian Anomalies

Table 2 lists the details of sixcases with obstructive Müllerian anomalies who suffered from endometriosis. Two of the six patients had primary amenorrhea. For the remaining four cases, the age at menarche ranged from 10 to 13 years and the age at surgery ranged from 12 to 23 years. The shortest and the longest interval between menarche and surgery was 2 years and 12 years, respectively. For two cases with primary amenorrhea, the interval between the beginning of symptoms (cyclic colic pain) and surgery was 3 years and 6 months, respectively. Of the six patients, two patients were diagnosed with HWW syndrome with unilateral cervical hypoplasia, two patients were diagnosed with unilateral non-communicating functional uterine horn, and two patients were diagnosed with cervical aplasia and upper vaginal defect. The rASRM score ranged from 1 to 96 (median 52.5), and five out of six cases were stage III or IV. Three of the six patients (50.0%) had endometrioma and four of the six patients (66.7%) had hematosalpinx. The postoperative follow-up period ranged from 6 to 138 months. Only one out of six cases had postoperative hormonal therapy, which was gonadotropin releasing hormone agonist (GnRHa) followed by low-dose estrogen–progestin (LEP). One out of five cases had endometrioma after outflow release surgery, and the recurrence rate of endometriosis was 20.0%. In case 6, the patient did not undergo surgery for release of the obstructive outflow.

### 3.3. Comparison between Endometriosis Patients under 24 Years Old with or without Obstructive Müllerian Anomaly

Next, we compared the patients’ backgrounds between endometriosis patients with obstructive Müllerian anomalies (*n* = 6) and endometriosis patients aged under 24 years without obstructive Müllerian anomalies (*n* = 31). Of the 37 total endometriosis patients aged under 24 years old, 6 patients were co-occurring with Müllerian anomalies (6/37, 16.2%). Of the 5 endometriosis patients aged under 19 years, 4 patients were complicated with Müllerian anomalies (4/5, 80.0%). All endometriosis patients without obstructive Müllerian anomalies underwent laparoscopic cystectomy, and the rASRM score ranged from 21 to 104. As shown in Table 3, there was no difference between the two groups regarding parity, BMI, and rASRM score. The age at surgery was significantly younger in endometriosis patients with obstructive Müllerian anomalies, compared with endometriosis patients without obstructive Müllerian anomalies (17.8 ± 4.4 vs. 23.1 ± 1.3, *p* = 0.0007). The rate of endometrioma was 50.0% in the group of endometriosis patients with obstructive Müllerian anomalies, and the rate of hydrosalpinx was significantly higher in the group of endometriosis patients with obstructive Müllerian anomalies (66.7% vs. 0%, *p* = 0.0002). The postoperative follow-up period was comparable between the two groups. For the group of endometriosis patients without obstructive Müllerian anomalies, 26 out of 31 patients (83.9%) performed postoperative hormonal treatment, which was significantly higher than that of the group of endometriosis patients with obstructive Müllerian anomalies. The recurrence rate of endometriosis was high as 25.9% (7/27) in the group of endometriosis patients without obstructive Müllerian anomalies, although there were no significant differences between the two groups. Of the endometriosis patients without obstructive Müllerian anomalies, three out of six patients without or discontinuous of postoperative hormonal treatment had endometrioma recurrence (50.0%), and four out of twenty-one patients with postoperative hormonal treatment had endometrioma recurrence (19.0%).

## 4. Discussion

In our study, 6 out of 12 patients with obstructive Müllerian anomalies developed endometriosis. When compared with endometriosis patients aged under 24 years old without Müllerian anomalies, the rate of endometrioma was 50.0% in the group of endometriosis patients with obstructive Müllerian anomalies, and the rate of hydrosalpinx was significantly higher in the group of endometriosis patients with obstructive Müllerian anomalies. Further, the recurrence rate of endometriosis was 20.0% in the group of endometriosis patients with obstructive Müllerian anomalies, whereas the recurrence rate of endometriosis was 25.9% in group of endometriosis patients without obstructive Müllerian anomalies.

It is known that retrograde menstruation causes endometriosis. Some reports revealed that the association between endometriosis and the obstruction of menstrual outflow caused by Müllerian anomalies. In our study, 6 out of 12 cases with obstructive Müllerian anomalies developed endometriosis. These six cases included two cases of HWW syndrome with cervical hypoplasia, two cases of non-communicating functional uterine horn, and two cases of cervical aplasia. In the previous report analyzing HWW syndrome, the occurrence of pelvic endometriosis was 19.15% (18/94) [9]. In another retrospective study with 87 patients of HWW syndrome, endometriosis was found in 12/87 patients (13.8%): in 10 cases on the peritoneum, in 5 cases on the ovary and in 1 case each on the fallopian tube and diaphragm [10]. A systematic review of 734 OHVIRA cases reported that endometriosis was found in 13.6% of selected cases who underwent laparotomy or laparoscopy [11]. Pelvic endometriosis was significantly more frequent in patients with complete hemivaginal obstructions, and all of the ovarian endometrioma was ipsilateral to the vaginal septum [9]. In our study, two cases of HWW syndrome co-occurring with cervical hypoplasia suffered from endometriosis, and one of them had endometrioma ipsilateral to the affected side, which was consistent with the past report. For non-communicating functional uterine horn, one study reported that pelvic endometriosis was observed in 5/10 cases (50.0%), 3 being stage I and 2 stage II [12]. For cervical aplasia, one study reported that 7 out of 12 patients (58.3%) had stage I or II endometriosis. Another retrospective study of 96 patients revealed that 54 (56.0%) had pelvic endometriosis, with 41 cases present on the ovary and in 13 cases on the peritoneum [13]. A delay from first symptoms to surgery of more than 1 year had a higher incidence of endometriosis. In our study, the interval from first symptoms or menarche to surgery ranged from 0.5 to 9 years, which was relatively long. Early diagnosis and surgery are necessary to prevent endometriosis.

We compared the patients’ background between endometriosis patients with obstructive Müllerian anomalies and endometriosis patients aged under 24 years without obstructive Müllerian anomalies. Endometriosis patients with obstructive Müllerian anomalies were significantly younger compared with the patients without anomalies. For young patients with endometriosis, we should consider the possibility of complication of congenital structural abnormalities. Further, adolescent patients with severe dysmenorrhea may have Müllerian anomalies, and early medical intervention should be performed. It is not well known about pathogenesis of endometriosis in adolescents. The prevalence of adolescent endometriosis is quite rare, and it is unclear whether etiology differs between adolescents and adult patients. Other risk factors for endometriosis in adolescence than obstructive Müllerian anomalies is positive family history, early menarche, lean body size, low birth weight, intense physical activity and passive smoking increase, which is not quite similar to endometriosis in adults [7]. Although retrograde menstruation is the main theory of the development of endometriosis, there are other mechanisms like coelomic metaplasia, implantation through the blood or lymphatic vessels, or differentiation from blood cells, hormonal regulation of inflammatory responses, and failure of the immune system [2,14,15,16]. Neonatal uterine bleeding (NUB) is one of the explanations for endometriosis developing in adolescents. NUB is triggered by the rapid fall in circulating progesterone levels in the first few days after birth, and it affects 5% of newborn girls [17]. This endometrial breakdown and bleeding represent an important risk factor for early-onset endometriosis [17]. Second, increased platelet activation, especially in the context of preeclamptic pregnancies, and a more naive immune system, may further increase the implantation density of endometrial stem/progenitor cells at NUB compared to cyclic menstruation [17]. Adolescent endometriosis is typically characterized by red hemorrhagic peritoneal lesions and, in severe cases, by large ovarian endometriomas. A literature review of recent studies shows that a substantial proportion of girls with early-onset endometriosis have a severe (rASRM stage III or IV) disease at laparoscopy and the severity is largely caused by the presence of ovarian endometriomas [17]. In our study, all of the patients with endometrioma in adolescence without obstructive Müllerian anomalies had and the severe endometriosis (rASRM stage III or IV). On the other hand, only 50% of the patients of endometriosis with obstructive Müllerian anomalies have endometrioma, but the rate of severe endometriosis is relatively high (83.3%). Endometriosis patients with obstructive Müllerian anomalies are more likely accompanied by hamatosalpinx than the patients without anomalies. In the past reports, hematosalpinx was diagnosed in 9/87 patients of HWW syndrome (10.3%) [10] and in 4/10 patients of non-communicating uterine horn (40.0%) [12]. It is reasonable that the blockade of menstrual outflow in obstructive Müllerian anomalies easily causes hematocolpos and hematosalpinx. This difference in clinical characteristics suggests that pathogenesis of endometriosis differs between adolescent patients with and without obstructive Müllerian anomalies.

In our study, the recurrence rate of endometriosis was 20.0% in the group of patients with endometriosis and with obstructive Müllerian anomalies. Although there were no significant differences, the recurrence rate of endometriosis was high at 25.9% in the group of patients with endometriosis without obstructive Müllerian anomalies, in spite of a higher rate of using post-operative hormonal treatment (83.9%). Younger endometriosis patients need to be followed up for longer periods, and post-operative management is crucial. In general, post-operative hormonal treatment could reduce the recurrence rate [18,19]. However, in our study, the recurrence of endometriosis in younger patients with hormonal treatment is comparable to that of adult patients without hormonal treatment. It is not known whether early-onset endometriosis tends to have a higher recurrence. One retrospective cohort study reported that the recurrence rate for endometriomas after surgical excisions at laparoscopy in adolescence is 36.84% [20]. Surgery is an effective way of treating endometriosis, but it also has disadvantages in adolescence. It increases the risk of premature ovarian failure caused by surgical treatment of ovarian endometriomas. It is unclear whether early operative intervention is the best way to reduce the recurrence for young endometriosis patients. Clinicians are recommended to adopt a personalized approach in the decision-making process to identify the best option for young women with endometriosis, after considering the age, symptoms, complication of obstructive Müllerian anomalies and the plan for childbearing. For patients with obstructive Müllerian anomalies, one in five patients had a recurrence of endometriosis. Regarding the course of endometriosis lesions after release of obstruction, there are either reports that it would be spontaneously refreshed or that it would persist after surgery [21,22,23,24]. Endometriosis that complicates congenital structural abnormality has the potential for difficulty in recurrence, as compared with young patients with endometriosis which is not accompanied by structural abnormality, once the outflow obstruction is released. Whether to perform intraperitoneal observation at the time of surgery or whether to perform postoperative hormonal therapy would be a future study. It is suggested that juvenile endometriosis patients have some kind of factor that uterine malformation patients do not have [7], and it is interesting to consider the pathogenesis mechanism of endometriosis. In uterine malformations, a few genes have garnered strong evidence of causality, mainly in syndromic presentations (HNF1B, WNT4, WNT7A, HOXA13) [25]. Among these genes, genome wide association studies (GWAS) identified common variants in locus near WNT4 [26]. This can represent a shared risk factor for endometriosis as well as uterine anomalies [5]. Other possible genetic background includes anatomical differences like the diameter of the cervix and the fallopian tube and functions of the uterine smooth muscles, which affect the outflow of menstrual blood. Further studies are required to detect more genetic susceptibility of early-onset endometriosis without obstructive Müllerian anomalies.

There are several limitations in our study. Firstly, this study is retrospective and preliminary due to the small number of participants. Secondly, not all patients with obstructive Müllerian anomalies underwent laparoscopic evaluation. The presence of endometriosis can be underestimated in those patients. Thirdly, we included only patients with endometriomas without obstructive Müllerian anomalies as a control which may not represent typical characteristics of adolescent endometriosis patients. Further larger clinical cohort studies are required for better understanding of the disease.

## 5. Conclusions

Endometriosis develops in half of adolescent patients with obstructive Müllerian anomalies. The clinical characteristics are different from patients without anomalies. Endometriosis with obstructive Müllerian anomalies is co-occurring with hematosalpinx more frequently than in patients without anomalies.

## Figures and Tables

**Table 1 biomedicines-12-00651-t001:** Twelve patients performed primary surgery for obstructive Müllerian anomalies: patients’ diagnosis and surgical management.

Diagnosis	Anatomic Morphology	Number of Cases	Surgeries	Endometriosis Co-Occurence Rate
HWW Syndrome without cervical abnormality	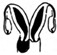	6 (5: noncommunicating vagina)	Vaginal Septum resection without inspection laparoscopy	N/A
HWW Syndrome with unilateral cervical hypoplasia	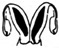	2	laparoscopic unilateral supra-cervical hysterectomy	2/2 (100%)
Non-communicating functional uterine horn	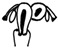	2	laparoscopic uterine horn removal	2/2 (100%)
Cervical aplasia	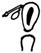	2	1: plastic surgery to vagina1: inspection laparoscopy	2/2 (100%)

HWW: Herlyn–Werner–Wunderlich; N/A: No endometrioima or hematosalpinx on MRI.

**Table 2 biomedicines-12-00651-t002:** Six patients with endometriosis: patients’ malformation type and endometriosis details.

Case	1	2	3	4	5	6
Malformation type	HHW syndrome with left cervical hypoplasia	HHW syndrome with left cervical hypoplasia	Left non-communicating functional uterine horn	Left non-communicating functional uterine horn	cervical aplasia with upper vaginal defect	Right unicorn uterus + cervical aplasia + upper vaginal defect + Left non-functional uterine horn
Age at Menarche (years)	10	13	10	13	primary amenorrhea	primary amenorrhea
Age at surgery (years)	12	23	19	19	21	13
interval between first symptom or menarche to a surgery (years)	2	10	9	6	0.5	3
Operation type	laparoscopic left supra hysterectomy	laparoscopic left supra hysterectomy	laparoscopic uterine horn removal	laparoscopic uterine horn removal	plastic surgery to vagina	inspection laparoscopy
rASRM score (stage)	1 (I)	83 (IV)	69 (IV)	96 (IV)	34 (III)	36 (III)
Endometrioma	−	−	−	+(left 6 cm)	+(right 1 cm, left 1.5 cm)	+(right 5 cm)
Hematosalpinx	−	+(left)	+(left)	+(left)	−	+(left)
postoperative follow-up period	27 months	6 months	16 months	33 months	138 months	29 months
postoperative hormonal therapy	−	−	−	−	−	GnRHa → LEP
Endometriosis recurrence	−	−	−	+(at 19 month)	−	N/A

HWW: Herlyn–Werner–Wunderlich, GnRHa: Gonadotropin releasing hormone agonist, LEP: low-dose estrogen–progestin.

**Table 3 biomedicines-12-00651-t003:** Comparison between endometriosis patients under 24 years old with or without obstructive Müllerian anomaly.

	Endometriosis with Müllerian Anomaly (*n* = 6)	Endometriosis without Müllerian Anomaly (*n* = 31)	*p*-Value
age at surgery (mean ± SD)	17.8 ± 4.4	23.1 ± 1.3	** 0.0002 **
19 years or younger (%)	66.7 (4/6)	3.2 (1/31)	** 0.0011 **
non parity (%)	100 (0/6)	100 (0/31)	1.00
BMI [median (range)]	19.3 (17.0–21.8)	19.5 (17.2–28.2)	0.53
rASRM score [median (range)]	52.5 (1–96)	48.5 (21–104)	0.90
rASRM stage (III–IV) (%)	83.3 (5/6)	100 (31/31)	0.16
endometrioma (%)	50 (3/6)	100 (31/31)	
hematosalpinx (%)	66.7 (4/6)	0 (0/31)	** 0.0002 **
postoperative medication (%)	16.7 (1/6)	83.9 (26/31)	** 0.003 **
	GnRHa → LEP 1	LEP 22	
		dienogest 1	
		LEP → dienogest 3	
follow-up period [months, median (range)]	28 (6–138)	37 (1–109)	0.64
total recurrence rate (patients followed up for at least 6 months) (%)	20.0 (1/5)	25.9 (7/27)	1.00
recurrence rate without or discontinuous of hormonal treatment (patients followed up for at least 6 months) (%)	20.0 (1/5)	50.0 (3/6)	0.55

GnRHa: Gonadotropin releasing hormone agonist, LEP: low-dose estrogen–progestin data were compared using Fisher’s exact test and Mann–Whitney test. A *p*-value of less than 0.05 was considered statistically significant, and all reported *p*-values were one-sided. A *p*-value of less than 0.05 was considered statistically significant and marked with red bold date font.

## Data Availability

Data is unavailable due to privacy restrictions.

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
