# Peer review of "The Association between Endometriosis and Obstructive Müllerian Anomalies"

_biomedicines, 2024, doi:10.3390/biomedicines12030651_

Round 1
Reviewer 1 Report
Comments and Suggestions for Authors
The authors investigated the characteristics of endometriosis patients with obstructive Müllerian anomalies. They compared those patients with patients with ovarian endometrioma aged under 24 years old. Although this study attempted to clarify the characteristics of endometriosis in obstructive Müllerian anomalies, there are many problems with this study.
1. First, the authors should clearly state the aims of this study in the introduction. It had been described that endometriosis patients with obstructive Müllerian abnormalities are compared with endometriosis patients without Müllerian abnormalities, but it is necessary to clearly describe what the comparison is intended to reveal.
2. They should select appropriate controls that meet their aims. They compared patients with obstructive Müllerian anomalies to patients with ovarian endometrioma, but the purposes of surgery were also different in these groups, and meaningful comparative data is limited. For example, it is inevitable that the frequency of ovarian endometriosis will be 100% in the control group because it is a case selection criterion, and there is no point in comparing it with the control group.
3. They should describe in their methods that there is no bias in their case selection criteria. In other words, the total number of each surgical procedure performed on patients under 24 years of age during the research period should be clearly described, and the exclusion criteria should be shown.
4. In this study, it is unclear how many patients with endometriosis are under 24 years old without developing ovarian endometrioma. However, those were included in the endometriosis group with Müllerian anomalies, and age comparisons between these two groups are meaningless considering the period it takes for lesions to form.
5. Although they concluded that Müllerian anomalies should be considered in young patients with severe endometriosis, their data was based on evaluation of severity on laparoscopic findings. This cannot be said that Müllerian anomalies will be discovered based on the findings. If they attempt to say that the degree of endometriosis suggests an anomaly, data such as symptoms should be compared.
Author Response
Reviewer1
The authors investigated the characteristics of endometriosis patients with obstructive Müllerian anomalies. They compared those patients with patients with ovarian endometrioma aged under 24 years old. Although this study attempted to clarify the characteristics of endometriosis in obstructive Müllerian anomalies, there are many problems with this study.
- First, the authors should clearly state the aims of this study in the introduction. It had been described that endometriosis patients with obstructive Müllerian abnormalities are compared with endometriosis patients without Müllerian abnormalities, but it is necessary to clearly describe what the comparison is intended to reveal.
Thank you for your comment. We state the aim of this study in the introduction part (Line 56-58).
- They should select appropriate controls that meet their aims. They compared patients with obstructive Müllerian anomalies to patients with ovarian endometrioma, but the purposes of surgery were also different in these groups, and meaningful comparative data is limited. For example, it is inevitable that the frequency of ovarian endometriosis will be 100% in the control group because it is a case selection criterion, and there is no point in comparing it with the control group.
As you indicated, control patients in our study may not represent typical characteristics of endometriosis patients in younger ages. However, we selected patients with ovarian endometrioma as control in order to accurately diagnose endometriosis and compared detailed intraoperative evaluation. We removed statistical analysis for the comparison of rate of endometrioma in the text (Line 151-152) and in figure 3. We mentioned this limitation in the discussion part (Line 281-283).
- They should describe in their methods that there is no bias in their case selection criteria. In other words, the total number of each surgical procedure performed on patients under 24 years of age during the research period should be clearly described, and the exclusion criteria should be shown.
We clearly state the inclusion criteria in the revised manuscript (Line 61-72). Among 131 patients aged under 24 years old who underwent surgeries for benign gynecological diseases in this period, patients who underwent primary surgery for obstructive Müllerian anomalies and patients who underwent endometrioma sized over 6 cm were included in this study. Two patients were excluded because of the previous operation history. Remaining 43 patients were analyzed in this study. We included all the patients who underwent surgery for patients with obstructive Müllerian anomalies and patients with over 6 cm endometriomas in this period. There is no selection bias in our study.
- In this study, it is unclear how many patients with endometriosis are under 24 years old without developing ovarian endometrioma. However, those were included in the endometriosis group with Müllerian anomalies, and age comparisons between these two groups are meaningless considering the period it takes for lesions to form.
Eighty-three patients under 24 years old were diagnosed with endometriosis in our outpatient hospital in this period, which is not overlapping with patients with Müllerian anomalies. However, endometriosis patients who did not have over 6 cm endometrioma did not undergo surgery and the diagnosis is based on the symptoms and pelvic exams, which is not as accurate as laparoscopy. Our study only included patients who underwent laparoscopic surgery in order to compare intraoperative findings accurately. For patients Mullerian anomalies, we did not include patients who did not undergo laparoscopic surgery as well although they may have endometriosis lesions. We wrote this limitation in the discussion (279-283).
- Although they concluded that Müllerian anomalies should be considered in young patients with severe endometriosis, their data was based on evaluation of severity on laparoscopic findings. This cannot be said that Müllerian anomalies will be discovered based on the findings. If they attempt to say that the degree of endometriosis suggests an anomaly, data such as symptoms should be compared.
Thank you for your comment. As you indicated, not all severe adolescence patients are cooccurring Müllerian anomalies and we cannot conclude this based on our findings. We removed the statement you indicated in conclusion. However, based on our findings, patients with anomalies are easy to develop endometriosis and most of them are severe. Besides, they cooccurred with hematosalpinx more frequently compared with patients without anomalies. This indicates the difference in pathogenesis between adolescence patients with or without anomalies. In addition, based on the higher recurrence rate of endometriosis in young patients without anomalies, they may have the different genetic backgrounds for the development of endometriosis. We believe our study provides useful information for physicians when treating obstructive Müllerian anomalies as well as young endometriosis patients. We simplify the result interpretation and change the statement in conclusion (Line 286-289). In the future study, we would like to perform larger cohort study to compare symptoms as well.
Reviewer 2 Report
Comments and Suggestions for Authors
General comments
The manuscript is interesting and raises the important problem of the co-occurrence of uterine malformations and endometriosis. The introduction is concise and clearly written. The aim of the study is well-defined, although it could be made more specific as an assessment of the co-occurrence of uterine malformations and endometriosis based on demographic data, clinical classification, and follow-up monitoring.
My main doubts are in the materials and methods section. This section requires significant enlargement. I suggest dividing it into three subsections 2.1. Study design, 2.2. Data collection, 2.3. Data analysis or statistical analysis.
The first subsection should contain information from L 61 - 69 extended with a detailed description of the inclusion and exclusion criteria for the study, including the size of the groups (see detailed comments below).
The second subsection should contain information from L 69 - 75 extended with a detailed description of obtaining data for research. The collection of demographic data such as age at menarche and age at surgery as well as basic clinical data such as type of surgery, and the interval between the first symptom or menarche to surgery do not raise any doubts. Examination protocols (clinical examination, US, and MRI) should be thoroughly described here and supported by appropriate references. Next, the American Society for Reproductive Medicine (rASRM) classification requires a broader copy supported by appropriate references. In addition, it is advisable to provide diagnostic criteria for endometrioma, hematosalpinx, endometriosis, and obstructive Müllerian anomalies so that the reader has no doubts about how the patients were diagnosed. How the follow-up period was monitored (what data was collected, how often, and how), including postoperative hormonal treatment, also requires a broader description.
The third subsection should contain information from L 80 - 83 extended with a detailed description of the acquired data - continuous or discrete data. Then, tests for compliance of the distribution of continuous data with the normal distribution and the results of these tests should be described. Then you should present what was compared with what and with what tests concerning the distribution of the compared data. Only such a subsection will be possible to assess whether the statistical analysis was carried out correctly or not. Consider whether Chi-square analysis would be appropriate to compare the data presented in Table 1 and partially in Table 2.
Detailed comments
L 47-48 This sentence needs minor rewording to make it clear that uterine malformations are a primary replacement that may be complicated by endometriosis.
L 48 Remove "M" after "Matalliotakis"
L 50, L 119, L 248 Remove additional space mark
L 57 There is no need to enter the number of cases to work, especially since you are also analyzing 31 cases regarding ovarian endometrioma.
L 68 How many patients were excluded for this reason and how many remained for further analysis?
L 75 Any abbreviation should be expanded when it is first used, even MRI.
L 93, L 97 Uses consistently previously entered abbreviations (MRI vs. MR imaging).
L 95 I appreciate the figures attached to Table 1. Although this is a non-standard solution, I think it is very good.
L 98 Consider replacing "complicated" with "co-occurring" here and throughout the manuscript.
L 106 Consider using "co-occurring" endometriosis.
L 135 lower instead of "younger"?
L 137, L 139, and L 144 "rate of ..." how was the rate calculated? as a percentage? occurrence? it should be stated in the M&M section.
L 140 "The postoperative follow-up period was comparable between two groups." Were these two groups any different than the two groups previously described? if so, what and why? this should also be covered in the M&M section.
L 152 Change the outer lines of Table 3 from thick to thin. In the p-value column, round the values to two decimal places where possible. Where it is not, round the values to the first digit other than 0. If you want to emphasize the significance of the differences, use a red font color and describe it in the table caption. For example, "A p-value of less than 0.05 was considered statistically significant and marked with bold font".
L 152 In Table 3 change "Endometrioma" to "endometrioma".
L 156 The Discussion section is well written following a standard format. However, it lacks a reference to a slightly broader reference to the pathogenesis of endometriosis. For this purpose, I can recommend two good review papers to the authors: 10.3390/ijms24032901 and 10.7150/ijbs.72707. Both of them refer to hormonal regulation in the pathogenesis of endometrosis, which, as I see it, is within the authors' interests.
L 265 The first and most important (not third) limitation is the small number of cases analyzed, which makes these studies preliminary. Given the other limitations noted, the M&M section does not provide information on how many patients were evaluated on the complete protocol and how many on the partial protocol. Such data should be included in the M&M section and clearly explained there, rather than just mentioned here in the limitation paragraph.
L 273 Unfortunately, the conclusions need to be reworded. Please look at the proposed changes to the thesis objective and reword the conclusions so that they answer the question posed in the thesis objective based on the results you obtained. Avoid all speculations.
Author Response
Reviewer2
The manuscript is interesting and raises the important problem of the co-occurrence of uterine malformations and endometriosis. The introduction is concise and clearly written. The aim of the study is well-defined, although it could be made more specific as an assessment of the co-occurrence of uterine malformations and endometriosis based on demographic data, clinical classification, and follow-up monitoring.
My main doubts are in the materials and methods section. This section requires significant enlargement. I suggest dividing it into three subsections 2.1. Study design, 2.2. Data collection, 2.3. Data analysis or statistical analysis.
The first subsection should contain information from L 61 - 69 extended with a detailed description of the inclusion and exclusion criteria for the study, including the size of the groups (see detailed comments below).
We clearly describe inclusion and exclusion criteria (61-72).
The second subsection should contain information from L 69 - 75 extended with a detailed description of obtaining data for research. The collection of demographic data such as age at menarche and age at surgery as well as basic clinical data such as type of surgery, and the interval between the first symptom or menarche to surgery do not raise any doubts. Examination protocols (clinical examination, US, and MRI) should be thoroughly described here and supported by appropriate references. Next, the American Society for Reproductive Medicine (rASRM) classification requires a broader copy supported by appropriate references. In addition, it is advisable to provide diagnostic criteria for endometrioma, hematosalpinx, endometriosis, and obstructive Müllerian anomalies so that the reader has no doubts about how the patients were diagnosed. How the follow-up period was monitored (what data was collected, how often, and how), including postoperative hormonal treatment, also requires a broader description.
Preoperative diagnosis of endometrioma and Müllerian anomalies was performed based on pelvic exam, US and MRI. Müllerian anomalies are complicated, so we show the pictures of anomalies in each case in table 1 for the readers to easily understand the anomalies. The presence of endometriosis and hematosalpinx were confirmed by laparoscopic evaluation. The reference for rASRM is added. Follow up interval was also added. As your suggestion, we changed the M&M sections in the revised manuscript (Line 81-87).
The third subsection should contain information from L 80 - 83 extended with a detailed description of the acquired data - continuous or discrete data. Then, tests for compliance of the distribution of continuous data with the normal distribution and the results of these tests should be described. Then you should present what was compared with what and with what tests concerning the distribution of the compared data. Only such a subsection will be possible to assess whether the statistical analysis was carried out correctly or not. Consider whether Chi-square analysis would be appropriate to compare the data presented in Table 1 and partially in Table 2.
Our data does not show normal distributions based on Shapiro-Wilk test and Kolmogorov-Smirnov test. We used Fisher’s exact test and Wilcoxon rank sum test for non-parametric analysis. We revised part of statistical analysis according to your suggestion (Line 92-98).
Detailed comments
L 47-48 This sentence needs minor rewording to make it clear that uterine malformations are a primary replacement that may be complicated by endometriosis.
Thank you for your comment. We reworded the sentence you indicated (Line 48-49).
L 48 Remove "M" after "Matalliotakis"
We removed M.
L 50, L 119, L 248 Remove additional space mark
We removed additional space mark.
L 57 There is no need to enter the number of cases to work, especially since you are also analyzing 31 cases regarding ovarian endometrioma.
We changed this part and state the aim of study clearly (Line 56-58). Thank you for your comment.
L 68 How many patients were excluded for this reason and how many remained for further analysis?
We clearly state the inclusion criteria (Line 61-72). Among 131 patients aged under 24 years old who underwent surgeries for benign gynecological diseases in this period, patients who underwent primary surgery for obstructive Müllerian anomalies and patients who underwent endometrioma sized over 6 cm were included in this study. Two patients were excluded because of the previous operation history. Remaining 43 patients were analyzed in this study.
L 75 Any abbreviation should be expanded when it is first used, even MRI.
We correctly used abbreviation in the revised manuscript. Thank you.
L 93, L 97 Uses consistently previously entered abbreviations (MRI vs. MR imaging).
We used MRI instead of MR imaging.
L 95 I appreciate the figures attached to Table 1. Although this is a non-standard solution, I think it is very good.
Thank you for your comment.
L 98 Consider replacing "complicated" with "co-occurring" here and throughout the manuscript.
We used co-occurring throughout the text.
L 106 Consider using "co-occurring" endometriosis.
We used co-occurring.
L 135 lower instead of "younger"?
I think younger is right.
L 137, L 139, and L 144 "rate of ..." how was the rate calculated? as a percentage? occurrence? it should be stated in the M&M section.
The rate was calculated based on the number of patients who diagnosed endometriosis in our study. Six patients with anomalies and 31 patients without anomalies are diagnosed with endometriosis. We stated the information of age and the number of patients included in this study in the M&M sections (Line 67-71).
L 140 "The postoperative follow-up period was comparable between two groups." Were these two groups any different than the two groups previously described? if so, what and why? this should also be covered in the M&M section.
This is the same as the previously described.
L 152 Change the outer Lines of Table 3 from thick to thin. In the p-value column, round the values to two decimal places where possible. Where it is not, round the values to the first digit other than 0. If you want to emphasize the significance of the differences, use a red font color and describe it in the table caption. For example, "A p-value of less than 0.05 was considered statistically significant and marked with bold font".
We changed Table 3 as your suggestion. Thank you.
L 152 In Table 3 change "Endometrioma" to "endometrioma".
We changed Endometrioma to endometrioma in Table 3.
L 156 The Discussion section is well written following a standard format. However, it lacks a reference to a slightly broader reference to the pathogenesis of endometriosis. For this purpose, I can recommend two good review papers to the authors: 10.3390/ijms24032901 and 10.7150/ijbs.72707. Both of them refer to hormonal regulation in the pathogenesis of endometrosis, which, as I see it, is within the authors' interests.
We added these references in discussion (Line 220-221, ref 15-16). Thank you.
L 265 The first and most important (not third) limitation is the small number of cases analyzed, which makes these studies preliminary. Given the other limitations noted, the M&M section does not provide information on how many patients were evaluated on the complete protocol and how many on the partial protocol. Such data should be included in the M&M section and clearly explained there, rather than just mentioned here in the limitation paragraph.
Thank you for your comment. The small number of sizes is the limitation of this study. We put it in the first part of limitation section (Line 278-279). We also clearly described the inclusion criteria in M&M (Line 61-72).
L 273 Unfortunately, the conclusions need to be reworded. Please look at the proposed changes to the thesis objective and reword the conclusions so that they answer the question posed in the thesis objective based on the results you obtained. Avoid all speculations.
Thank you for your comment. We reworded the sentences in conclusion and avoid any speculations. We tried to simplify the results interpretation and conclusion (Line 286-289). Our findings are that patients with anomalies tend to easily develop endometriosis and cooccurring with hematosalpinx compared with patients without anomalies. This indicates the difference in pathogenesis between adolescence patients with or without anomalies. In addition, based on the higher recurrence rate of endometriosis in young patients without anomalies, they may have the different genetic backgrounds for the development of endometriosis. We believe our study provides useful information for physicians when treating obstructive Müllerian anomalies as well as young endometriosis patients.
Round 2
Reviewer 1 Report
Comments and Suggestions for Authors
The authors have almost successfully revised the manuscript in accordance with previous comments. Case selection criteria were well described, and conclusions have also been revised to match the results. However, there remains still non-significant statistical data. Because the selection criteria for the target group and control group are different, there is no significance in calculating the proportion of Müllerian anomaly cases relative to the combined population (in line 191-192). This data is misleading to readers and the authors should revise the description.
Author Response
Thank you for your comment. As you indicated, the proportion of obstructive Mullerian anomalies in endometriosis patients in our study does not reflect actual proportion. We removed the sentence you mentioned.
Reviewer 2 Report
Comments and Suggestions for Authors
The authors put a lot of work into improving the manuscript. I appreciate their effort. However, I still have a few minor comments that should be addressed before the paper is accepted for publication.
First, please include in the M&M section the subsections I requested in the first round of reviews.
Dividing the uniform M&M text into three subsections (2.1. Study design, 2.2. Data collection, 2.3. Statistical analysis) will increase its readability.
Can you explain why you used two tests examining the distribution of the data? Note that the Shapiro-Wilk test is recommended over the Kolmogorov-Smirnov test. Was the test result always the same? What if he wasn't? I recommend that you use only one test (the Shapiro-Wilk test) and use its results to select subsequent tests. There is no information about the result of the normal distribution test for the tested data series. Were all data series distributed normally or some not?
The Wilcoxon rank sum test is a test for paired data in which at least one data series is not normally distributed. Without information about the result of the normal distribution test, I cannot assess whether this test was selected correctly. Additionally, I am not convinced that the paired data test selection is appropriate for comparing Patients' characteristics between patients with or without obstructive Müllerian anomaly. After all, they weren't the same patients. Consider whether a Mann-Whitney test would be more appropriate (if at least one data series was not normally distributed) or simply an Unpaired t-test with Welch's correction (if both data series were normally distributed but had different SD).
Having clarified these minor remarks, I am pleased to say that the article can be accepted for publication.
Author Response
The authors put a lot of work into improving the manuscript. I appreciate their effort. However, I still have a few minor comments that should be addressed before the paper is accepted for publication.
First, please include in the M&M section the subsections I requested in the first round of reviews.
Dividing the uniform M&M text into three subsections (2.1. Study design, 2.2. Data collection, 2.3. Statistical analysis) will increase its readability.
Thank you for your suggestion. M&M section is divided into three categories.
Can you explain why you used two tests examining the distribution of the data? Note that the Shapiro-Wilk test is recommended over the Kolmogorov-Smirnov test. Was the test result always the same? What if he wasn't? I recommend that you use only one test (the Shapiro-Wilk test) and use its results to select subsequent tests. There is no information about the result of the normal distribution test for the tested data series. Were all data series distributed normally or some not?
We used Shapiro-Wilk test as your suggestion, and all data does not have normal distribution. Thank you for your useful advice.
The Wilcoxon rank sum test is a test for paired data in which at least one data series is not normally distributed. Without information about the result of the normal distribution test, I cannot assess whether this test was selected correctly. Additionally, I am not convinced that the paired data test selection is appropriate for comparing Patients' characteristics between patients with or without obstructive Müllerian anomaly. After all, they weren't the same patients. Consider whether a Mann-Whitney test would be more appropriate (if at least one data series was not normally distributed) or simply an Unpaired t-test with Welch's correction (if both data series were normally distributed but had different SD).
Thank you for your comment. You’re right, and Mann-Whitney test is appropriate. We re-analyzed the data with Mann-Whitney. We slightly changed p-value in figure 6.
Having clarified these minor remarks, I am pleased to say that the article can be accepted for publication.
I appreciate your review. Thank you.